# Interpretable machine learning for cardiovascular risk prediction: Insights from NHANES dietary and health data

**Md Ahiduzzaman**[1]*, **Md Nahid Hasan**[2]

1 Department of Statistics and Data Science, University of Central Florida, Orlando, Florida, United States of America, 2 Department of Mathematics, East Texas A&M University, Commerce, Texas, United States of America

☯ These authors contributed equally to this work.

* Md.Ahiduzzaman@ucf.edu

## Abstract

**Background:** Cardiovascular diseases (CVD) are one of the leading global causes of death, which requires an accurate early prediction. This study aimed to develop transparent machine learning (ML) models using National Health and Nutrition Examination Survey (NHANES) data from 2017–2023 to predict CVD risk based on dietary and health factors.

**Methods:** We analyzed data from 12,382 adults (aged 18 and older) from NHANES 2017–2023, including 41 dietary, anthropometric, clinical, and demographic variables. Recursive Feature Elimination (RFE) was used to select an optimal subset of 30 predictors. To address substantial class imbalance in the outcome, we applied the Random Over-Sampling Examples (ROSE) technique to the training data. Five machine learning models—Logistic Regression, Random Forest, Support Vector Machines, XGBoost, and LightGBM—were trained and evaluated. Model interpretability was assessed using LIME and SHAP.

**Results:** Participants with CVD differed significantly from those without CVD in age, waist circumference, systolic blood pressure, C-reactive protein (CRP), and multiple dietary nutrients, with a consistently lower nutrient intake in the CVD group. Among the ML models evaluated, XGBoost achieved the highest accuracy (0.8216) and recall (0.8645), while Random Forest showed the highest AUROC (0.8139). Interpretability analyses identified age as the strongest predictor, followed by vitamin B12, total cholesterol, CRP, and waist circumference.

**Conclusion:** Interpretable ML models effectively identified key dietary and clinical factors for CVD risk. Nutrients like vitamin B12 and niacin, alongside established clinical indicators, emerged as significant predictors, underscoring their potential role in nutritional interventions and public health strategies for CVD prevention.

**Data availability statement:** The data used in this study are publicly available from the National Health and Nutrition Examination Survey (NHANES) website at https://www.cdc.gov/nchs/nhanes. The raw data is also available at (https://www.kaggle.com/datasets/ahiduzzaman28/nhanes-cvd-raw-data-2017-23).

**Funding:** The author(s) received no specific funding for this work.

**Competing interests:** No authors have competing interests.

## Introduction

Cardiovascular disease (CVD) refers to a group of disorders that affect the heart and blood vessels, including coronary artery disease, cerebrovascular disease, and peripheral arterial disease. These conditions can impair the cardiovascular system's ability to circulate blood efficiently and regulate vascular function, often leading to serious health outcomes such as coronary heart disease, congestive heart failure, stroke, or premature death. Due to its complex and multifactorial nature—driven by behavioral, metabolic, environmental, and genetic factors—CVD remains a leading cause of morbidity and mortality worldwide. In recent years, machine learning (ML) approaches have gained momentum in CVD research for their capacity to analyze high-dimensional health data and improve early risk prediction and classification.

Several studies have applied ML to predict CVD and related outcomes using integrated health data. Recently, Mao et al. [1] reviewed the application of the ML algorithm in the diagnosis of heart diseases. Klados et al. [2] developed a machine learning framework to predict CVD based on physical functioning and health status, while Ngamdu et al. [3] found that advanced periodontal disease was strongly associated with higher CVD risk. Dinh et al. [4] employed ensemble models to identify key predictors of CVD and diabetes. More broadly, ML has shown promise in modeling a variety of health outcomes beyond cardiovascular disease. For instance, recent studies have applied ML techniques to predict conditions such as asthma and insomnia [5,6]. These findings highlight the capacity of ML to uncover complex, multidomain relationships in population health—particularly when applied to large-scale, nationally representative datasets that include clinical, dietary, and biomarker information.

One such resource is the National Health and Nutrition Examination Survey (NHANES), a nationally representative dataset widely used to study cardiovascular disease, particularly in identifying and analyzing its risk factors. Prior studies have highlighted the predictive value of various biomarkers and exposures, such as the FT3/FT4 thyroid hormone ratio [7], heavy metals like cadmium and lead [8], and trends in hypertension and lipid treatment across obesity levels [9]. Other work has linked micronutrients like vitamin D and cryptoxanthin to reduced inflammation and oxidative stress [10,11], while caffeine intake has shown mixed associations depending on dosage [12]. NHANES has also supported research on comorbidities and disparities, including the link between COPD and CVD [13], racial/ethnic differences in cardiometabolic outcomes [14], and the role of dietary data [15].

However, most studies to date have often examined individual risk factors independently, without fully utilizing the diverse types of health information available across domains. To address this gap, we developed an integrated ML framework using the most recent NHANES data from 2017 to 2023. We incorporate macro- and micronutrient intake, demographics, laboratory biomarkers, anthropometric measures, and clinical examination data to predict CVD risk. Multiple ML classifiers—including logistic regression (LR), random forests (RF), support vector machines (SVM), XGBoost, and LightGBM—are evaluated to identify optimal performance.

We further apply SHapley Additive exPlanations (SHAP) and Local Interpretable Model-agnostic Explanations (LIME) to enhance model transparency and provide insights into the most influential predictors.

This research contributes to the growing field of data-driven CVD prevention by demonstrating how machine learning can effectively integrate diverse health data while providing interpretable results that could inform targeted public health interventions.

## Materials and methods

### Data source and study population

This study utilized data from the National Health and Nutrition Examination Survey (NHANES) spanning 2017–2023, encompassing both pre-pandemic (2017–2020) and post-pandemic (2021–2023) periods [16]. NHANES is a nationally representative survey conducted by the National Center for Health Statistics that combines interviews, physical examinations, and laboratory tests to assess the health and nutritional status of the U.S. population.

The analysis included 12,382 adults aged 18 years and older after data preprocessing. We integrated multiple NHANES components: dietary intake data, demographic information, anthropometric measurements, clinical examinations, laboratory biomarkers, and self-reported medical conditions.

### Outcome variable

The primary outcome was a binary indicator of cardiovascular disease (CVD) based on self-reported physician diagnoses. Participants were classified as having CVD (coded as 1) if they reported ever being diagnosed with angina, congestive heart failure, coronary heart disease, heart attack, or stroke. These conditions are recognized as key indicators of cardiovascular disease [17]. Those reporting none of these conditions were classified as CVD-free (coded as 0). The classification process is summarized in Table 1.

The co-occurrence matrix in Fig 1 illustrates the frequency with which different cardiovascular diseases occur together within the dataset, providing insights into potential comorbidities and patterns of disease co-existence among individuals.

### Predictor variables

In this study, we selected 41 features across five key domains: Dietary nutrients: Macronutrients (protein, carbohydrates, fats), vitamins (B6, B12, C, D, E, K, thiamin, riboflavin, niacin, folate), minerals (calcium, iron, magnesium, zinc, copper, potassium, sodium, selenium), and other compounds (beta-carotene, lutein, fiber); Anthropometric measures: Body Mass Index (BMI) and waist circumference; Clinical markers: Systolic and diastolic blood pressure; Laboratory biomarkers: Total cholesterol and C-reactive protein; and Demographic factors: Age.

### Data preprocessing

We implemented a systematic strategy to address missing data, perform feature selection, and handle class imbalance prior to model development. The preprocessing workflow is illustrated in Fig 2.

**Table 1. Classifying CVD based on self-reported health conditions.**

| Question | CVD status | Assigned label |
|---|---|---|
| Responded "yes" to having had one of the following diseases[1] | Have cardiovascular diseases | 1 |
| Responded "no" to all [1] | Have no cardiovascular diseases | 0 |

[1] Angina, Coronary Heart Disease, Congestive Heart Failure, Heart Attack, and Stroke.

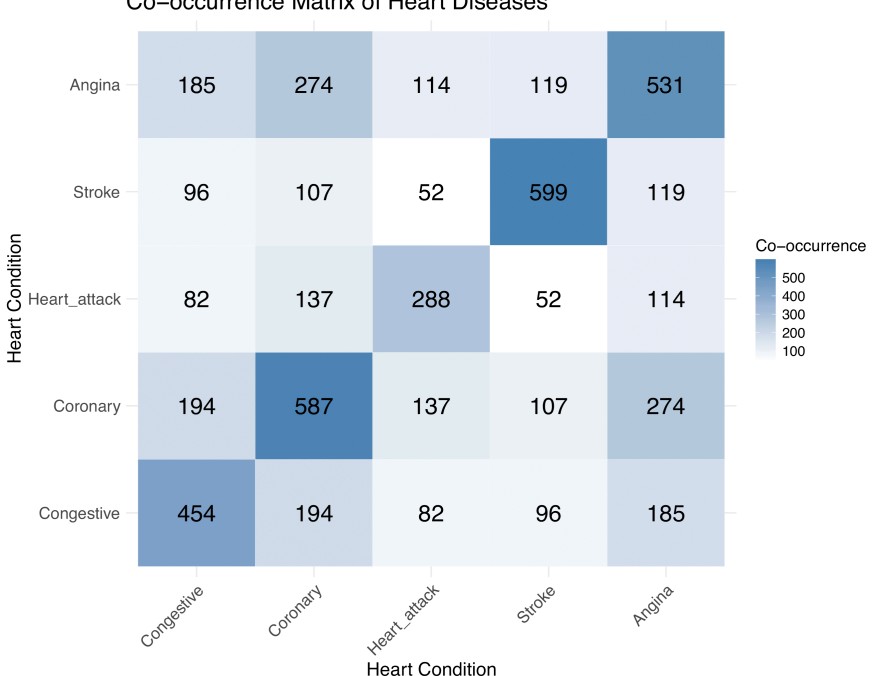

**Fig 1**. **Co-occurrence matrix of cardiovascular diseases, illustrating the frequency of co-existing conditions in the dataset.**

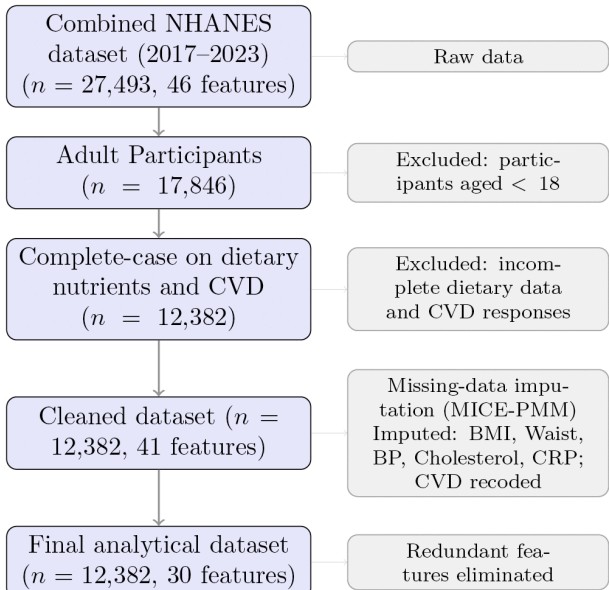

**Fig 2**. **Data preprocessing workflow from raw data to modeling-ready dataset.**

Initially, we addressed missing data through a two-stage approach. First, we performed complete-case analysis for dietary variables and the outcome-CVD. We observed uniform missingness in dietary variables, which is approximately

30.6%. Second, for remaining variables with lower missingness rates (1.14%–7.13%), we applied Multiple Imputation by Chained Equations (MICE) [18] using Predictive Mean Matching, generating five imputed datasets with 50 iterations each.

Next, we applied feature selection to enhance model accuracy and interpretability. Recursive Feature Elimination (RFE) with random forest as the base estimator was used, incorporating 5-fold cross-validation repeated five times. This process identified 30 optimal features that contributed most to predictive performance. The selected variables covered all original domains and included important predictors such as age, total cholesterol, waist circumference, blood pressure, and key vitamins and minerals. A justification plot is given in the result section Fig 3.

The final analytical dataset comprised 12,382 participants with 30 carefully selected features, providing a robust foundation for cardiovascular disease prediction modeling. Details is shown in Table 2.

## Modeling approach

The final analytical dataset was first split into training (80%) and testing (20%) subsets. Due to the substantial class imbalance in the outcome variable (with 87.7% (10,864) non-CVD cases 12.3% (1,518) CVD cases ), we applied the Random Over-Sampling Examples (ROSE) technique [19] to the training data, keeping the test set untouched. This approach synthetically balanced the classes, resulting in a training set with 4,592 CVD and 4,695 non-CVD cases. Using this balanced training set, we trained five machine learning classifiers: Logistic Regression (LR) [20], Random Forests (RF) [21], Support Vector Machines (SVM) [22], Extreme Gradient Boosting (XGBoost) [23], and Light Gradient Boosting Machine (LightGBM) [24]. Finally, model performance was evaluated on the test set using metrics including Accuracy, Precision, Recall, Specificity, F1 Score, and Area Under the ROC Curve (AUROC).

To interpret contributions of each feature to the model, we employed Local Interpretable Model-agnostic Explanations (LIME) [25] and SHapley Additive exPlanations (SHAP) [26]. LIME provided insights into individual predictions, while SHAP analyses facilitated a detailed decomposition of global and local feature influences, enhancing both interpretability and model transparency. SHAP visualizations were generated to intuitively demonstrate the relative importance of features on model predictions.

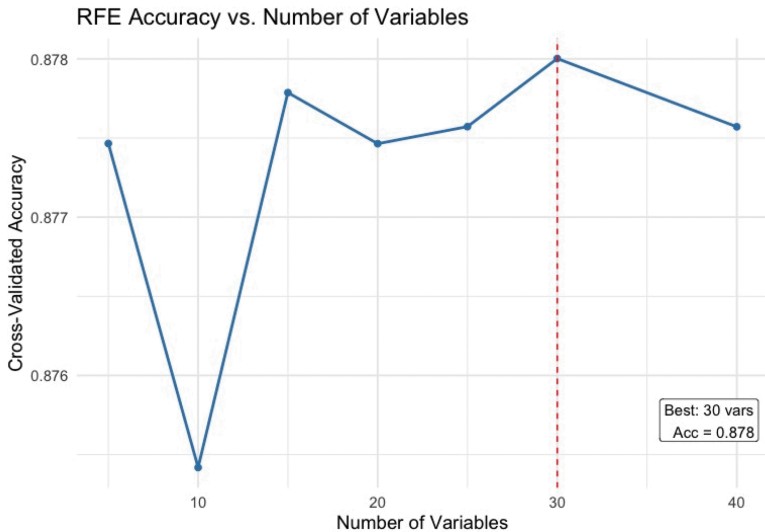

**Fig 3**. **RFE plot for feature selection.**

**Table 2. Descriptive summary of final dataset with *p*-values.**

| Characteristic | Class 0 (*N* = 10,864)[1] | Class 1 (*N* = 1,518)[1] | *p*-value[2] |
|---|---|---|---|
| Protein | 78 (42) | 71 (39) | <0.001 |
| Carbohydrates | 239 (123) | 220 (120) | <0.001 |
| Sugars | 102 (74) | 98 (72) | 0.048 |
| Fiber | 17 (11) | 15 (10) | <0.001 |
| Saturated Fat | 27 (17) | 26 (18) | 0.002 |
| Monounsaturated Fat | 29 (18) | 28 (17) | <0.001 |
| Polyunsaturated Fat | 20 (14) | 19 (12) | <0.001 |
| Cholesterol | 314 (259) | 303 (253) | 0.047 |
| Beta Carotene | 2,340 (4,297) | 2,150 (4,137) | <0.001 |
| Cryptoxanthin | 91 (236) | 92 (205) | 0.300 |
| Lutein Zeaxanthin | 1,545 (3,107) | 1,408 (3,019) | <0.001 |
| Thiamin | 1.50 (0.87) | 1.42 (0.82) | <0.001 |
| Riboflavin | 1.92 (1.26) | 1.83 (1.24) | 0.002 |
| Niacin | 25 (17) | 22 (13) | <0.001 |
| Vitamin B6 | 1.98 (1.85) | 1.77 (1.43) | <0.001 |
| Folic Acid | 144 (150) | 132 (136) | 0.001 |
| Food Folate | 212 (143) | 189 (131) | <0.001 |
| Iron | 456 (311) | 413 (280) | <0.001 |
| Choline | 331 (203) | 318 (206) | 0.003 |
| Vitamin B12 | 4.5 (6.0) | 4.4 (11.3) | 0.092 |
| Vitamin C | 80 (92) | 76 (97) | 0.004 |
| Vitamin D | 4.4 (5.5) | 4.6 (5.2) | 0.001 |
| Vitamin E | 9.3 (7.0) | 8.4 (6.1) | <0.001 |
| Vitamin K | 124 (172) | 108 (164) | <0.001 |
| Calcium | 898 (562) | 850 (561) | <0.001 |
| Phosphorus | 1,314 (677) | 1,222 (665) | <0.001 |
| Magnesium | 296 (155) | 270 (148) | <0.001 |
| Zinc | 10.2 (7.9) | 9.5 (5.9) | <0.001 |
| Copper | 1.18 (1.00) | 1.10 (2.00) | <0.001 |
| Sodium | 3,326 (1,804) | 3,022 (1,671) | <0.001 |
| Potassium | 2,518 (1,274) | 2,386 (1,268) | <0.001 |
| Selenium | 110 (66) | 102 (64) | <0.001 |
| Moisture | 2,929 (1,503) | 2,660 (1,484) | <0.001 |
| Age | 50 (17) | 66 (12) | <0.001 |
| BMI | 30 (7) | 31 (8) | <0.001 |
| Waist Circumference | 101 (17) | 108 (17) | <0.001 |
| Systolic BP | 123 (18) | 130 (23) | <0.001 |
| Diastolic BP | 75 (11) | 74 (13) | 0.002 |
| Total Cholesterol | 189 (41) | 171 (43) | <0.001 |
| C-Reactive Protein | 3.9 (7.2) | 5.0 (9.4) | <0.001 |

[1]Mean (SD)   [2]Mann-Whitney *U* Test

## Model specifications

To ensure reproducibility, we report the exact settings used for each classifier. We did not run an extensive hyperparameter search; instead we used literature-guided defaults with early stopping where applicable, applied to the *training* split only (ROSE-balanced; the test set was untouched). The SVM (`e1071`) used a radial basis function (RBF) kernel with cost ($C = 1$) and $\gamma = 1/p$ (package default; $p = 30$ predictors after RFE), with `probability=TRUE` to obtain class probabilities. XGBoost (`xgboost`) was fit with `objective=binary:logistic`, `eval_metric=auc`, `max_depth=8`, `eta=0.1`, `subsample=0.8`, `colsample_bytree=0.8`, `min_child_weight=1`, `scale_pos_weight=1`, up to 1000 rounds with early stopping at 50 rounds. Random Forest (`randomForest`) used `ntree=500` with default ($mtry = \lfloor\sqrt{p}\rfloor$). Logistic regression (`stats::glm`) used a binomial logit link with all selected predictors. LightGBM (`lightgbm`) was trained

with `objective=binary`, `metric=auc`, `learning_rate`=0.01, `num_leaves`=31, and `scale_pos_weight` set to the inverse class ratio in the training split, with early stopping at 50 rounds. All models used the same 30-feature set selected by RFE and fixed random seeds.

### Ethics statement

The data used in this study were obtained from the National Health and Nutrition Examination Survey (NHANES), a publicly available dataset provided by the National Center for Health Statistics (NCHS). NHANES is conducted in compliance with the ethical guidelines established by the NCHS, and all participants provided informed consent at the time of data collection. This study utilized de-identified publicly available data. The analysis adhered to all applicable ethical standards for research using secondary data.

## Results

### Descriptive statistics

Table 2 summarizes the dietary, demographic, anthropometric, and clinical characteristics of the study population, stratified by CVD status. Significant group differences (with p-values <0.05) were found in key demographic and clinical measures, including age, waist circumference, systolic blood pressure, and C-reactive protein. Participants diagnosed with CVD (Class 1) showed significantly lower mean dietary intake of nutrients, including protein, carbohydrates, sugars, fiber, vitamins B6, C, K, iron, zinc, and magnesium compared to participants without CVD (Class 0). However, no significant differences were observed for cryptoxanthin (p-value = 0.30) and vitamin B12 (p-value = 0.092). The RFE plot in Fig 3 shows the performance stabilized after 15 predictors and peaked at 30 predictors with CV accuracy ≈ 0.878. We chose 30 features because the RFE curve peaked at 30 and this subset retains most of the nutrient profile alongside key clinical CVD markers—whereas a 15-feature subset would discard many nutrition variables central to our research question.

### Performance of machine learning models

Table 3 summarizes the performance of five ML models evaluated. XGBoost offers the best balance of accuracy (0.8216) and recall (0.8645), which is essential for detecting high-risk individuals. LR provides the highest specificity (0.7810) and precision (0.9576), making it particularly effective at minimizing false-positive predictions. RF (0.8790) and LightGBM (0.8883) achieved strong F1-scores, reflecting balanced performance between precision and recall. Overall, the results suggest that ensemble-based models—particularly XGBoost and LightGBM—are well-suited for capturing complex patterns in the data and offer strong predictive performance for CVD risk classification.

Fig 4 compares the ROC curves for all classifiers and illustrates their relative ability to distinguish between CVD and non-CVD cases. RF achieved the highest AUROC (0.814), with XGBoost (0.808), LR and LightGBM (both 0.807), and SVM (0.798) showing comparable but slightly lower performance.

**Table 3. Performance ML models.**

| Metric | XGBoost | Random Forest | Logistic Regression | LightGBM | SVM |
|---|---|---|---|---|---|
| Accuracy | **0.8216** | 0.7994 | 0.7008 | 0.8120 | 0.7548 |
| AUROC | 0.8082 | **0.8139** | 0.8065 | 0.8066 | 0.7977 |
| Precision | 0.9273 | 0.9337 | **0.9576** | 0.9278 | 0.9402 |
| Recall | **0.8645** | 0.8303 | 0.6896 | 0.8520 | 0.7695 |
| Specificity | 0.5145 | 0.5778 | **0.7810** | 0.5251 | 0.6491 |
| F1-score | **0.8948** | 0.8790 | 0.8018 | 0.8883 | 0.8463 |

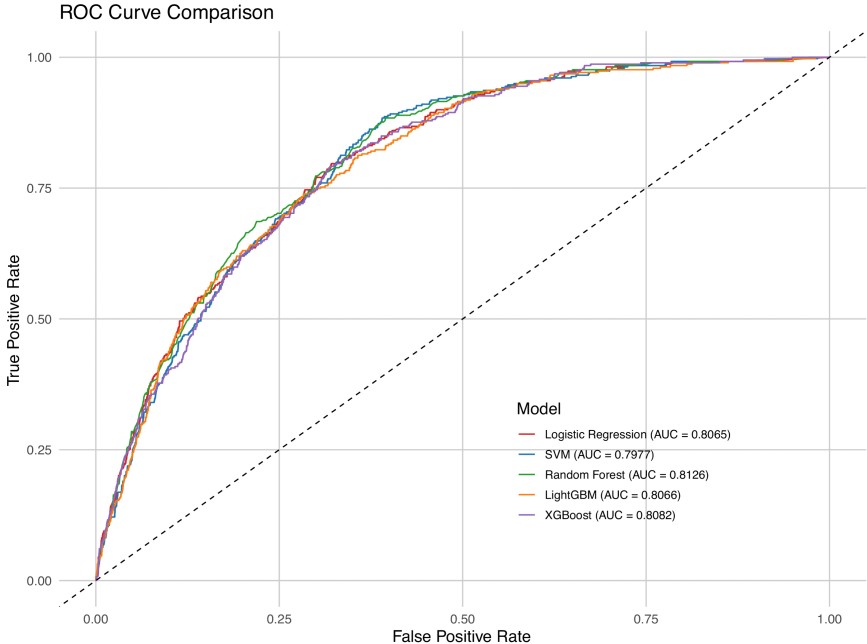

**Fig 4**. **ROC curve comparison for all ML models.**

## Feature importance and model interpretability

LIME and SHAP were used to assess feature importance and model interpretability, as shown in Fig 5 and Fig 6, respectively. LIME analyzes of individual predictions frequently highlighted age, vitamin B12, total cholesterol, C-reactive protein (CRP), waist circumference and niacin as key factors that influence predictions, with specific levels of nutrients influencing risk estimates.

SHAP analysis provided a comprehensive view of feature contributions across the dataset. Higher values of age and total cholesterol were consistently correlated with increased SHAP values, indicating higher predicted cardiovascular risk. In addition to clinical factors, dietary factors such as vitamin B12 and niacin also contribute significantly to the prediction of cardiovascular risk. These insights underscore the potential value of incorporating detailed nutrient-level data into predictive models for the assessment of CVD risk.

Interestingly, The SHAP beeswarm plot shows for the Age and Total cholesterol there might be a interaction effect. We have therefore assessed the assumption with a SHAP dependency plot. Fig 7 shows how Total_Cholesterol relates to the model's predicted CVD risk, with points colored by Age. Each point is one participant. The vertical axis shows whether cholesterol moved the prediction up (positive values) or down (negative values). Two simple patterns appear: (i) among older adults with low cholesterol, cholesterol often pushes risk up; and (ii) above about 180–200 mg/dL, cholesterol adds little extra signal once other strong factors (e.g., age, waist size, blood pressure, CRP) are already considered. This should not be read as "high cholesterol is protective"—it only means that its added contribution is small in those regions. The far right tail has few points, so those extremes should be interpreted cautiously. Medication use was not available and may influence the low-cholesterol/older cluster.

## Discussion

In this study, we developed and evaluated five ML models to predict a composite CVD outcome using the nationally representative dataset from NHANES 2017–2023. Our results indicated that the ensemble models, specifically XGBoost,

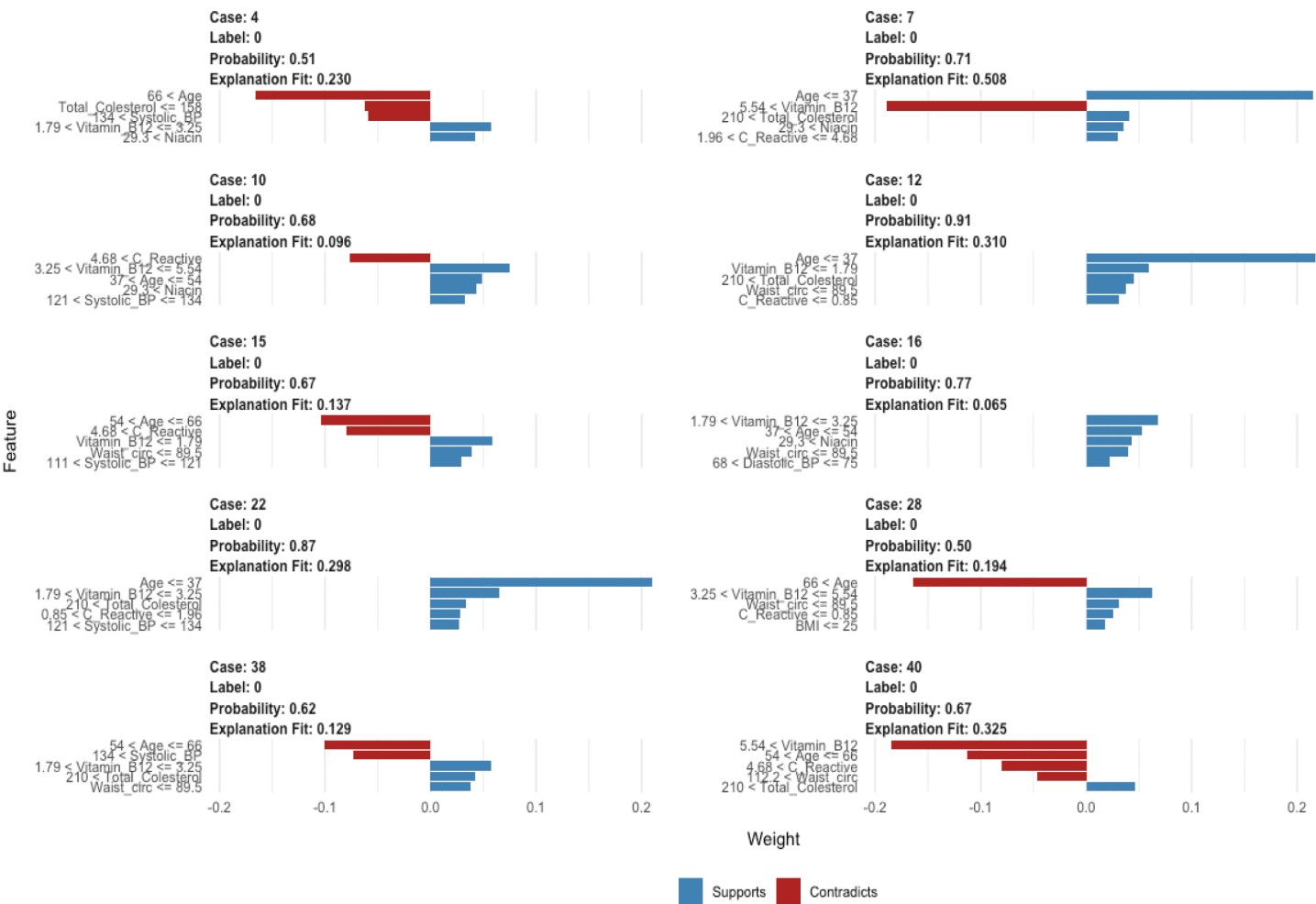

**Fig 5**. **LIME explanations for selected individual predictions, showing features supporting or contradicting the predictions.**

achieved the highest accuracy (0.8216) and recall (0.8645), while RF demonstrated the best overall discriminative performance with an AUROC of 0.8139.

The overall performance of XGBoost and RF of this study aligns with findings from broader literature. In a meta-analysis, Krittanawong et al. [27], indicated that boosting algorithms were promising for coronary artery disease prediction. In another study, Azmi et al. [28] and Naser et al. [29] also frequently reported RF as a high-performing algorithm in CVD prediction. However, a direct comparison of the performance of the ML models should be made with caution due to differences in the datasets and the definition of the outcome variable [30,31]. Our results remain robust and appropriate for the complexity of a large nationally representative survey.

In addition, this study demonstrates that integrating detailed dietary nutrient profiles with demographic and clinical variables can enhance the prediction of CVD risk. Interpretability analyses using LIME and SHAP confirmed the importance of both clinical and nutritional features—particularly age, CRP, total cholesterol, and vitamin B12—in shaping CVD risk predictions. For instance, SHAP values indicated how varying levels of these features, including specific nutrients, shifted the predicted risk. This in-depth perspective on the role of individual nutrients is a key outcome, as many studies on CVD prediction tend to focus on broader dietary patterns or a more limited set of biochemical markers rather than an extensive

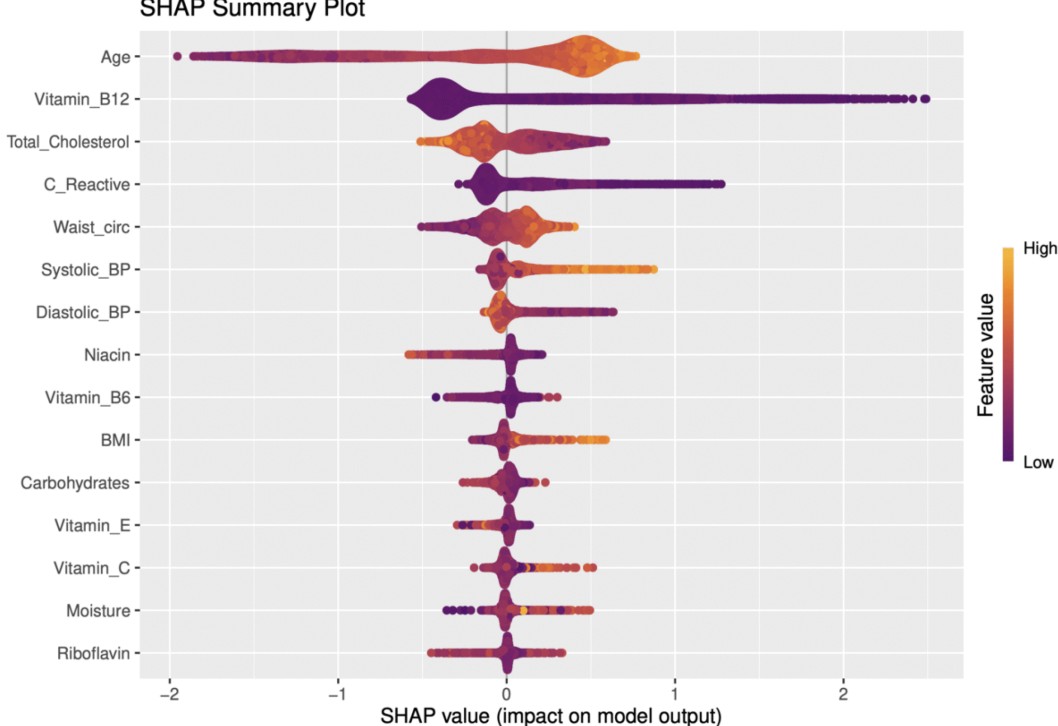

**Fig 6**. SHAP beeswarm plot illustrating the distribution and impact of feature values on model predictions.

profile of 26 dietary nutrients as explored here [28,30,31]. Our results highlight the importance of nutrients such as vitamin B12 and niacin in the risk of cardiovascular disease, aligning with recent findings by Yang et al. [32]. Further, demonstrating the value of interpretable machine learning methods to uncover specific associations between nutrient diseases.

A distinct strength of this research, beyond the detailed nutrient analysis, is the use of NHANES data from 2017–2023, which uniquely includes the post-pandemic period, a time frame largely unaddressed in the cited comparative literature [5,6,27,28]. This allows our findings to reflect potentially newer trends in population health and nutrition. The methodological rigor, including systematic handling of missing data (MICE), addressing class imbalance (ROSE)—a challenge also noted by Ogunpola et al. [30] and Naser et al. [29] —and robust feature selection using Recursive Feature Elimination (RFE) further strengthens our study's validity. At the global level, the SHAP summary plot showed that higher *Age*, larger *Waist_circ*, and elevated blood pressure increased predicted CVD risk, whereas lower values reduced it. *Total_Cholesterol* displayed a non-linear, age-dependent pattern: very low values in older adults tended to raise predicted risk, while values above ~180–200 mg/dL added little incremental signal once age, adiposity, blood pressure, and CRP were accounted for. LIME panels for borderline test cases clarified which features pushed an individual prediction up (blue) or down (red), providing person-level transparency.

From a public-health perspective, these patterns emphasize modifiable targets—central adiposity, blood pressure, and inflammation—alongside specific nutrient signals (e.g., vitamin B12, niacin). In practice, risk estimates could help prioritize counseling and hypertension control for higher-burden subgroups (e.g., older adults and those with central obesity); findings are associative rather than causal.

CVD status was derived from self-reported, physician-diagnosed conditions. While this approach is standard in population surveillance, it is subject to recall error and misclassification. We anticipate that any non-differential misclassification would reduce associations.

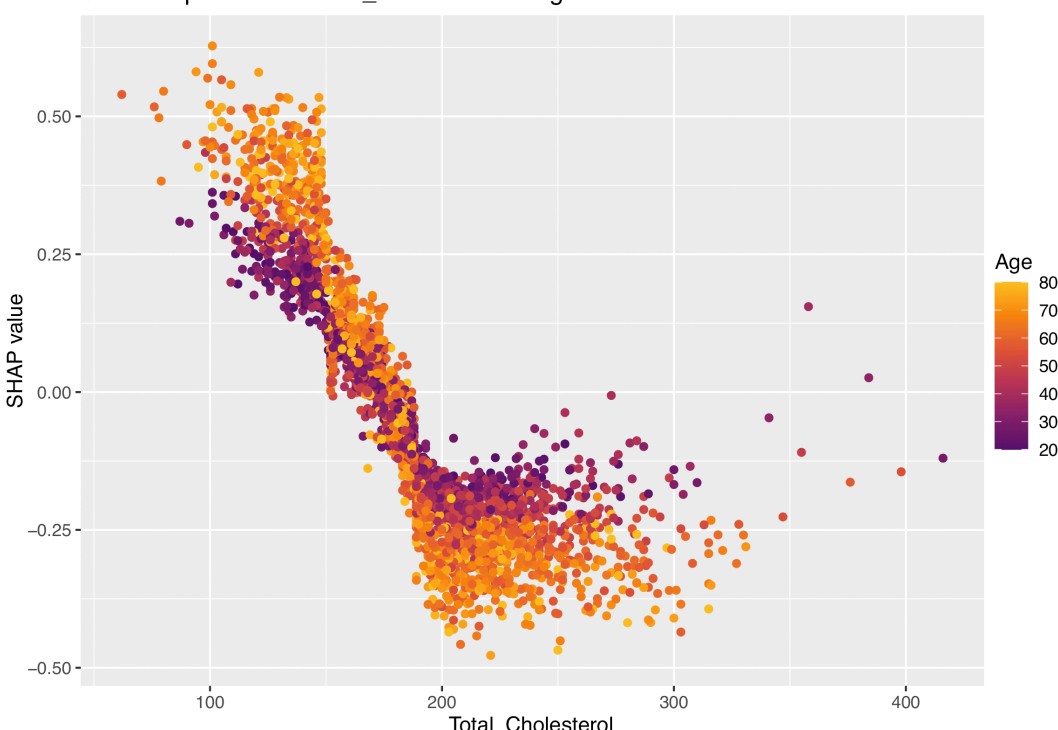

**Fig 7**. **SHAP dependence for Total_Cholesterol on the test set.**

Future research should aim to validate these findings using longitudinal data to better explore causal pathways, especially for identified nutritional predictors. Incorporating objectively measured CVD outcomes and further investigating the specific impact of the pandemic period on CVD risk factors are also important next steps. Stability analysis of model performance and feature importance, for instance, through bootstrap simulations as suggested by Huang et al. [33], would add another layer of robustness to the findings.

## Conclusion

Using a nationally representative NHANES sample spanning pre- and post-pandemic cycles, we applied explainable machine learning to a detailed profile of dietary and non-dietary variables and identified factors *associated* with prevalent CVD among U.S. adults. Model explanations (e.g., LIME and SHAP) and probability calibration analyses provide transparency about how predictors contribute to risk estimates within this dataset. These results are hypothesis-generating and may help prioritize variables and subgroups for further study rather than support individualized clinical decision-making at this stage.

It is important to note some limitations. First, CVD status was self-reported, which may introduce recall or classification error. Second, the cross-sectional design precludes causal inference; our findings should not be interpreted as evidence that modifying specific dietary components will reduce CVD risk. Third, 24-hour dietary recalls may not reflect long-term intake, and the complete-case approach for variables with substantial missingness reduced the analytic sample and may introduce selection bias. Finally, while model interpretability tools aid understanding, model complexity and potential dataset shift limit generalizability. External and temporal validation, incorporation of medication and additional clinical

biomarkers, and prospective evaluation of calibrated thresholds are important next steps before considering clinical application [33].

## Supporting information

**S1 File. Supplementary information.** Calibration curves before and after post-hoc calibration (S1 Fig); LIME explanations for the undersampled model (S2 Fig); and test-set discrimination and calibration metrics (S1 Table).
(PDF)

## Author contributions

**Data curation:** Md Ahiduzzaman.

**Investigation:** Md Ahiduzzaman.

**Methodology:** Md Ahiduzzaman, Md Nahid Hasan.

**Software:** Md Ahiduzzaman.

**Supervision:** Md Nahid Hasan.

**Validation:** Md Ahiduzzaman.

**Visualization:** Md Ahiduzzaman.

**Writing – original draft:** Md Ahiduzzaman.

**Writing – review & editing:** Md Ahiduzzaman, Md Nahid Hasan.

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
