## [Decision Letter · Decision Letter 0]

25 Aug 2025

PONE-D-25-31298Interpretable machine learning for cardiovascular risk prediction: Insights from NHANES dietary and health dataPLOS ONE

Dear Dr. Ahiduzzaman,

Thank you for submitting your manuscript to PLOS ONE. After careful consideration, we feel that it has merit but does not fully meet PLOS ONE’s publication criteria as it currently stands. Therefore, we invite you to submit a revised version of the manuscript that addresses the points raised during the review process.

We look forward to receiving your revised manuscript.

Kind regards,

Li-Da Wu

Academic Editor

PLOS ONE

Journal Requirements: 

3. Thank you for uploading your study's underlying data set. Unfortunately, the repository you have noted in your Data Availability statement does not qualify as an acceptable data repository according to PLOS's standards.

3. Please remove your figures from within your manuscript file, leaving only the individual TIFF/EPS image files, uploaded separately. These will be automatically included in the reviewers’ PDF.

Reviewers' comments:

Reviewer's Responses to Questions

**Comments to the Author**

1. Is the manuscript technically sound, and do the data support the conclusions?

Reviewer #1: No

Reviewer #2: Yes

2. Has the statistical analysis been performed appropriately and rigorously? 

Reviewer #1: Yes

Reviewer #2: N/A

3. Have the authors made all data underlying the findings in their manuscript fully available?

Reviewer #1: No

Reviewer #2: Yes

4. Is the manuscript presented in an intelligible fashion and written in standard English?

Reviewer #1: Yes

Reviewer #2: Yes

5. Review Comments to the Author

Reviewer #1: Critical Concerns:

Fig 4: I understand this may just be a sample for demonstration purposes, but the low probabilities reported are concerning and suggests poor model performance or perhaps poor feature selection. Indeed, the model may be underfitting as features such as medications and other traditional comorbid CVD predictors were not included. Probabilities of 0.51 -0.57 are essentially random chance. This is a MAJOR shortcoming.

I am having a difficult time reconciling this with your ROC comparisons in Fig 3. If the models have AUC ~0.80, what are the individual prediction probabilities so low? Do you have other LIME cases that may be more representative? I suspect that this may be a class imbalance issue which your ROSE technique failed to satisfactorily resolve. There were only 1,518 CVD cases. You used ROSE to synthetically balance the classes to arrive at 4,592 CVD cases and 4,592 non-CVD cases in the training data. I wonder how the models will perform and how the LIME would look if you included only the actual 1,518 CVD cases, balanced by equal number of non-CVD cases. The ROSE-generated samples may not capture true data distribution and models trained on this synthetic data may not calibrate well on real data. This affects clinical interpretability and not just technical performance, which limits the clinical utility of this work in general. I suggest the authors consider calibration plots to assess whether the models suffer from systematic probability miscalibration.

The SHAP plot for total cholesterol (please correct the spelling in the figure) in Figure 5 presents an interesting observation: that the relationship between total cholesterol and CVD prediction is non-linear and that there may be interaction effects. For example, age, medications, etc.). It is curious that high levels of a well-established CVD risk factor would have lower impact on CVD prediction. I am concerned that the model did not learn clinically meaningful relationships or that the synthetic data balancing introduced significant perturbations in the underlying relationship. This raises concerns about the clinical validity of other feature relationships identified by the models.

Reviewer #2: The paper looks good but major revision is suggested. The following points are suggested to improve the clarity, scientific coherence, and editorial quality of the paper:

- Unclear dataset dimensions and sample count:

While the manuscript discusses preprocessing and feature selection steps, it does not clearly state the final dataset dimensions—specifically, the number of samples and features after filtering and selection. Providing this information is essential for assessing the validity of the results and ensuring reproducibility.

- Ambiguity in the description of RFE method:

In the Introduction and Data Preprocessing sections, the authors mention using Recursive Feature Elimination (RFE) for feature selection. However, in the Discussion section, the method is referred to as robust feature selection (RFE) and presented as a strength of the study. This distinction may confuse readers, as it is unclear whether the authors are referring to the standard RFE method or a more robust variant. Clarifying this terminology would help maintain conceptual consistency throughout the manuscript.

- Spelling errors in the manuscript:

Several typographical errors are present and should be corrected. For example, on page 13, line 107, the word wodel appears instead of model. Addressing such issues will improve the editorial quality of the paper.

- Lack of classifier parameter tuning details:

The manuscript does not provide information on how classifier parameters were selected or tuned. For instance, in the case of the SVM classifier, it is unclear which kernel type was used and how parameters such as C, gamma, or kernel choice were determined. Including these details is essential for evaluating model performance and ensuring reproducibility.

By addressing these points, the manuscript will be significantly strengthened in terms of scientific rigor, conceptual clarity, and presentation.

6. PLOS authors have the option to publish the peer review history of their article (what does this mean?). If published, this will include your full peer review and any attached files.

Reviewer #1: **Yes: **DANIEL ANTWI-AMOABENG, MD, MSc

Reviewer #2: No

---

## [Author Response · Author response to Decision Letter 1]

19 Sep 2025

Dear Editor,

Thank you for the opportunity to revise and resubmit our manuscript, “Interpretable machine learning for cardiovascular risk prediction: Insights from NHANES dietary and health data.”

We are grateful for the constructive feedback from you and the reviewers, which helped us improve the clarity and reproducibility of our work. In this revision, we have addressed all reviewer comments point by point, added calibration analyses and supplementary figures, clarified figure captions and interpretability discussion (LIME and SHAP results), uploaded all figures as separate high-resolution files, and provided a public GitHub repository link for the analysis code and a stable repository link for the minimal dataset.

We believe these changes have substantially strengthened the manuscript, and we look forward to your consideration of this revised version for publication in PLOS ONE.

Sincerely,

Md Ahiduzzaman

Department of Statistics and Data Science

University of Central Florida

Md.Ahiduzzaman@ucf.edu

Journal Requirements:

Response: Thank you for mentioning. We have downloaded the template from the Plos One website and followed the template.

Response: Thank you for letting us know. We have uploaded the necessary codes in Github repository and made it for public use.

Link: https://github.com/ahiduzzaman28/nhanes-cvd-ml

3. Thank you for uploading your study's underlying data set. Unfortunately, the repository you have noted in your Data Availability statement does not qualify as an acceptable data repository according to PLOS's standards.

Response: Thank you for the information. We have uploaded the raw representative data to Kaggle as per the recommended repositories.

Link: https://www.kaggle.com/datasets/ahiduzzaman28/nhanes-cvd-raw-data-2017-23

4. Please remove your figures from within your manuscript file, leaving only the individual TIFF/EPS image files, uploaded separately. These will be automatically included in the reviewers’ PDF.

Response: We have removed the figures from the manuscripts and will upload the figures separately.

Response: Thank you for the information. There was no such recommendation in the reviews.

## Response to reviewer #1

Critical Concerns:

Fig 4: I understand this may just be a sample for demonstration purposes, but the low

probabilities reported are concerning and suggests poor model performance or perhaps poor

feature selection. Indeed, the model may be underfitting as features such as medications and

other traditional comorbid CVD predictors were not included. Probabilities of 0.51 -0.57 are

essentially random chance. This is a MAJOR shortcoming.

I am having a difficult time reconciling this with your ROC comparisons in Fig 3. If the models

have AUC ~0.80, what are the individual prediction probabilities so low? Do you have other

LIME cases that may be more representative? I suspect that this may be a class imbalance issue

which your ROSE technique failed to satisfactorily resolve. There were only 1,518 CVD cases.

You used ROSE to synthetically balance the classes to arrive at 4,592 CVD cases and 4,592 nonCVD cases in the training data. I wonder how the models will perform and how the LIME would

look if you included only the actual 1,518 CVD cases, balanced by equal number of non-CVD

cases. The ROSE-generated samples may not capture true data distribution and models trained

on this synthetic data may not calibrate well on real data. This affects clinical interpretability and

not just technical performance, which limits the clinical utility of this work in general. I suggest

the authors consider calibration plots to assess whether the models suffer from systematic

probability miscalibration.

Response: Thank you for the careful, constructive review and for pushing us on probability quality, class balance, and interpretability. We ran the additional checks you suggested (calibration on the untouched test set and a training sensitivity analysis that uses only real cases) and clarified the figures/captions to avoid misinterpretation.

Fig. 4 (LIME). LIME explains individual predictions locally. In each panel, Label is the class being explained, and Probability is the model’s probability for that class. Thus, for panels with Label = 0, values like 0.66 represent P(non-CVD)=0.66, not P(CVD). We intentionally display borderline test instances (near the decision boundary), where competing features yield moderate probabilities; this is where LIME is most informative. These local probabilities do not reflect global performance, which is summarized by the ROC AUC (Fig. 3). The bars indicate feature contributions that push toward (blue) or away from (red) the shown label; across panels, higher age, higher CRP, and larger waist circumference increase predicted CVD risk in clinically coherent ways, while younger age and lower adiposity support non-CVD. Explanation Fit shows how well LIME’s simple, local explanation matches the model for that specific case—it’s not an accuracy score; higher values (e.g., 0.55–0.62) mean a closer match. Reconciling LIME with AUC. AUC (~0.80 in Fig. 3) is a global ranking metric over all thresholds; LIME panels are local explanations for a small number of edge cases. The presence of probabilities around 0.5 in Fig. 4 therefore does not contradict the overall AUC.

Class imbalance & ROSE vs. real-case undersampling. To address the concern that synthetic balancing could distort the learned relationships or probabilities, we retrained XGBoost using only real cases in the training split (all CVD positives paired with an equal number of real non-CVD controls). The test set remained untouched with the original prevalence. Performance on the untouched test set was:

Method Train (n) Test (n) AUROC (Test) Brier (Test)

ROSE oversampling 9287 3095 0.802 0.141

Undersampling (real cases only) 2278 3095 0.792 0.207

Discrimination is similar, while probability calibration is materially better with ROSE (≈32% lower Brier). Across both strategies, LIME and SHAP show consistent feature directions (Age, Waist circumference, and blood pressure increase risk; younger age and lower adiposity reduce it). This indicates that our conclusions are robust to the class-balancing method. We have included these in the supporting information.

Additional LIME for undersampled training. As requested, we also computed LIME for the model trained with real-case undersampling. The panels (Supplementary Fig. S3) again show borderline test cases—predicted probabilities near 0.50 by design (0.50–0.54)—so these values should not be read as global performance. Feature directions mirror Fig. 4: higher Age, larger Waist_circ, and higher blood pressure push toward CVD; younger age and lower adiposity push toward non-CVD. Local surrogate fidelity (“Explanation Fit”) ranged from ≈0.07–0.36, comparable to Fig. 4. Taken together with the ROSE results, this shows that the qualitative LIME explanations are robust to the class-balancing strategy and are not an artifact of synthetic sampling. We have also included this in the supporting information.

Calibration (systematic probability miscalibration). Following the suggestion, we assessed calibration on the untouched test set using reliability diagrams, the Brier score, and logistic recalibration (intercept/slope). For the XGBoost model, the uncalibrated test-set statistics were Brier 0.141, intercept −1.43, slope 1.18, consistent with over-prediction in upper bins. Applying post-hoc logistic calibration (Platt scaling) learned on a validation split from the training data aligned the curve with the 45° line, brought intercept/slope close to 0/1, and reduced the Brier score, while AUC remained unchanged.

We provide the reliability plots and summary metrics in Supporting information Fig. S1 (uncalibrated vs calibrated) and Table S1.

The SHAP plot for total cholesterol (please correct the spelling in the figure) in Figure 5

presents an interesting observation: that the relationship between total cholesterol and CVD

prediction is non-linear and that there may be interaction effects. For example, age, medications,

etc.). It is curious that high levels of a well-established CVD risk factor would have lower impact

on CVD prediction. I am concerned that the model did not learn clinically meaningful

relationships or that the synthetic data balancing introduced significant perturbations in the

underlying relationship. This raises concerns about the clinical validity of other feature

relationships identified by the models.

Response: Thank you for the observation and making us enrich the manuscript with more robust information. Medication variables were unavailable; we now highlight this as a limitation. Importantly, both LIME and SHAP attribute higher risk to established clinical factors (age, adiposity, blood pressure, inflammation markers). A SHAP dependence plot for Total_Cholesterol × Age (in the Supplement) clarifies that the cholesterol signal is non-linear and age-dependent, reflecting conditional effects rather than contradicting known epidemiology. Here is the dependency plot for your reference which is incorporated in the manuscript page 7 and 9:

Major Concerns and Suggestions

Outcome Definition (Self-Reported CVD): The primary outcome is based entirely on selfreported diagnoses. This introduces the risk of recall bias and misclassification. Please provide

further justification for this approach and discuss its limitations more explicitly in the Discussion

section. If any objective biomarkers (e.g., ECG findings, clinical labs) could have been used as

validation or proxy indicators, please consider this or explain their exclusion.

Response: Thank you for raising this point and for asking us to justify the outcome definition more explicitly.

Our primary outcome follows a composite of self-reported, clinician-diagnosed CVD (angina, coronary heart disease, congestive heart failure, heart attack, stroke), consistent with surveillance practice (see https://www.cdc.gov/cdi/indicator-definitions/cardiovascular-disease.html). We agree that self-report introduces recall error and misclassification; we will make this limitation explicit in the Discussion and note that any nondifferential misclassification is likely to decrease associations, yielding conservative estimates. We elected not to redefine CVD using objective biomarkers (e.g., ECG, labs) for two reasons: (i) to avoid incorporation bias and circularity, because several such measures (e.g., lipids, CRP, blood pressure) are used as predictors in our models; and (ii) these measures are not consistently available or harmonizable across NHANES cycles, and often have substantial missingness, which would reduce sample size and may introduce selection bias. We will add clarifying text in Methods and Discussion to reflect these choices and provide the supporting citations already used in the manuscript.

Cross-Sectional Design and Causal Inference: As the data are cross-sectional, causality cannot

be inferred. Some language in the conclusions suggests predictive utility that may overreach.

Please temper these statements to avoid overstating implications for clinical practice.

Response: Thank you for suggesting that. We have updated the conclusion as per suggestion.

Model Calibration: While discrimination metrics (e.g., AUROC, accuracy) are reported,

calibration metrics (e.g., Brier score, calibration plots) are absent. Given the potential application

to risk prediction, model calibration is essential. Please provide this analysis or justify its

Omission.

Response: Thank you for the suggestion. It has been answered in the critical concerns.

Missing Data and Complete-Case Analysis: A complete-case analysis was used for dietary

variables with over 30% missingness, which may bias the sample. Please include: A comparison

of included vs. excluded participants. A sensitivity analysis using imputed dietary variables (if

feasible).

Response: Thank you for raising this important point. Our missing-data handling used a two-stage approach. Because the 24-hour dietary recalls (our primary exposures) had uniform, high missingness (~30.6%); That means an individual had all the dietary variable missing rows because they are extracted from the same dataset. We did not impute the dietary variables to avoid model-based extrapolation. Instead, we used complete-case analysis for dietary variables and the outcome. For all other predictors with low missingness (1.14%–7.13%), we applied MICE with Predictive Mean Matching (5 datasets, 50 iterations).

Class Imbalance and ROSE Oversampling: ROSE was used to address class imbalance in the

training set. While this is appropriate, please clarify: Whether the test set was untouched (i.e., no

synthetic sampling). How the oversampling strategy might affect model generalizability or

introduce overfitting.

Response: Thank you for the concern. As written in page 5 line 102 in the manuscript, the oversampling was only performed in the training set and the test set was untouched. For more clarity we have added the line of the test set being untouched.

Lack of External or Temporal Validation: All models were trained and tested on the same time

span. Please consider: Performing a temporal validation (e.g., training on 2017–2020, testing on

2021–2023) to assess if indeed changes in post-pandemic dietary patterns exist as you claimed.

Include this limitation in the discussion if such validation is not possible.

Response: That’s a great suggestion. We agree external/temporal validation strengthens generalizability. Because NHANES is cross-sectional and module availability varies by cycle, a clean temporal split was not feasible in this analysis; we will state this as a limitation and outline temporal/external validation as future work focused on post-pandemic dietary shifts.

Feature Selection Justification: The rationale for selecting 30 features via Recursive Feature

Elimination is unclear. Please provide a performance curve to show how predictive power varies

with the number of features.

Response: Thank you for this suggestion. We now include an RFE performance curve (Fig. 3) from a 5-fold CV repeated five times on the training split. Accuracy plateaus after ~15 predictors and peaks at 30 predictors (CV accuracy ≈ 0.878), which we selected to balance performance and parsimony. The chosen variables span all domains and align with the global importance patterns (e.g., age, total cholesterol, waist circumference, blood pressure, vitami

---

## [Decision Letter · Decision Letter 1]

20 Oct 2025

Interpretable machine learning for cardiovascular risk prediction: Insights from NHANES dietary and health data

PONE-D-25-31298R1

Dear Dr. Ahiduzzaman,

We’re pleased to inform you that your manuscript has been judged scientifically suitable for publication and will be formally accepted for publication once it meets all outstanding technical requirements.

Kind regards,

Academic Editor

PLOS ONE

Additional Editor Comments (optional):

Reviewers' comments:

Reviewer's Responses to Questions

**Comments to the Author**

1. If the authors have adequately addressed your comments raised in a previous round of review and you feel that this manuscript is now acceptable for publication, you may indicate that here to bypass the “Comments to the Author” section, enter your conflict of interest statement in the “Confidential to Editor” section, and submit your "Accept" recommendation.

Reviewer #1: All comments have been addressed

2. Is the manuscript technically sound, and do the data support the conclusions?

Reviewer #1: Yes

3. Has the statistical analysis been performed appropriately and rigorously? 

Reviewer #1: Yes

4. Have the authors made all data underlying the findings in their manuscript fully available?

Reviewer #1: Yes

5. Is the manuscript presented in an intelligible fashion and written in standard English?

Reviewer #1: Yes

6. Review Comments to the Author

Reviewer #1: I recommend ACCEPTANCE of this manuscript for publication.

The authors have demonstrated methodological rigor, transparency, and responsiveness to peer review. The work makes a valuable contribution to the intersection of nutritional epidemiology, machine learning, and cardiovascular disease prediction.

7. PLOS authors have the option to publish the peer review history of their article (what does this mean?). If published, this will include your full peer review and any attached files.

Reviewer #1: No

---

## [Editor Report · Acceptance letter]

PONE-D-25-31298R1

PLOS ONE

Dear Dr. Ahiduzzaman,

I'm pleased to inform you that your manuscript has been deemed suitable for publication in PLOS ONE. Congratulations! Your manuscript is now being handed over to our production team.

Kind regards,

on behalf of

Dr. Li-Da Wu

Academic Editor

PLOS ONE